# Strategies to identify individuals with monogenic diabetes: results of an economic evaluation

Jaime L Peters [ID] ,[1,2] Rob Anderson,[3] Beverley Shields,[4] Sophie King,[4] Michelle Hudson [ID] ,[4] Maggie Shepherd [ID] ,[4] Timothy James McDonald,[4,5] Ewan Pearson,[6] Andrew Hattersley,[4] Chris Hyde[1]

[1]Exeter Test Group, University of Exeter Medical School, Exeter, UK
[2]Collaboration for Leadership in Applied Health Research and Care South West Peninsula (NIHR CLAHRC South West Peninsula), University of Exeter Medical School, Exeter, UK
[3]ESMI (Evidence Synthesis & Modelling for Health Improvement), University of Exeter, Exeter, UK
[4]NIHR Clinical Research Facility, University of Exeter Medical School, Exeter, UK
[5]Royal Devon and Exeter NHS Foundation Trust, Exeter, UK
[6]Division of Molecular & Clinical Medicine, University of Dundee, Dundee, UK

**Correspondence to**
Dr Jaime L Peters;
j.peters@exeter.ac.uk

## ABSTRACT

**Objectives** To evaluate and compare the lifetime costs associated with strategies to identify individuals with monogenic diabetes and change their treatment to more appropriate therapy.

**Design** A decision analytical model from the perspective of the National Health Service (NHS) in England and Wales was developed and analysed. The model was informed by the literature, routinely collected data and a clinical study conducted in parallel with the modelling.

**Setting** Secondary care in the UK.

**Participants** Simulations based on characteristics of patients diagnosed with diabetes <30 years old.

**Interventions** Four test-treatment strategies to identify individuals with monogenic diabetes in a prevalent cohort of diabetics diagnosed under the age of 30 years were modelled: clinician-based genetic test referral, targeted genetic testing based on clinical prediction models, targeted genetic testing based on biomarkers, and blanket genetic testing. The results of the test-treatment strategies were compared with a strategy of no genetic testing.

**Primary and secondary outcome measures** Discounted lifetime costs, proportion of cases of monogenic diabetes identified.

**Results** Based on current evidence, strategies using clinical characteristics or biomarkers were estimated to save approximately £100–£200 per person with diabetes over a lifetime compared with no testing. Sensitivity analyses indicated that the prevalence of monogenic diabetes, the uptake of testing, and the frequency of home blood glucose monitoring had the largest impact on the results (ranging from savings of £400–£50 per person), but did not change the overall findings. The model is limited by many model inputs being based on very few individuals, and some long-term data informed by clinical opinion.

**Conclusions** Costs to the NHS could be saved with targeted genetic testing based on clinical characteristics or biomarkers. More research should focus on the economic case for the use of such strategies closer to the time of diabetes diagnosis.

**Trial registration number** NCT01238380.

## BACKGROUND

Monogenic diabetes is a form of diabetes caused by a mutation in a single gene, which

### Strengths and limitations of this study

► The model structure was informed by expert consultation and critical appraisal of existing models.
► Parameter values were taken from a UK-based clinical study conducted alongside this economic evaluation.
► Wide-ranging sensitivity analyses were conducted.
► Many parameters were based on low numbers of patients.
► Evidence on effectiveness was limited.

is inherited in an autosomal dominant manner.[1] Therefore a child of an individual with monogenic diabetes has a 50% chance of inheriting the mutation (assuming the child's other parent does not have the mutation). Mutations in glucokinase (*GCK*), hepatocyte nuclear factor 1 alpha (*HNF1A*) and hepatocyte nuclear factor 4 alpha (*HNF4A*) genes are the most common forms of monogenic diabetes.[2] Individuals with mutations in the *GCK gene* have persistently moderately raised blood glucose levels from birth, that is rarely detrimental to health[3] and does not respond to treatment.[4] Therefore individuals with mutations in the *GCK* gene can be successfully treated by diet.[4] Individuals with *HNF1A* or *HNF4A* mutations have blood glucose levels which increase over time and can be successfully treated with sulphonylureas[5] but may, eventually, require insulin treatment.[6]

The minimum prevalence of monogenic diabetes in the UK has been estimated as 108 cases per million.[7] As it usually presents by 25–30 years of age,[1 2 8] individuals are often misdiagnosed with type 1 diabetes, and receive insulin treatment when less invasive and less costly treatment is more appropriate.

The National Health Service (NHS) in England and Wales currently has no national guidelines for identifying individuals with monogenic diabetes. Realistic strategies are

available ranging from genetic testing of all individuals with diabetes to targeted genetic testing based on clinical characteristics[9] or biochemical[10] and immunological[11] tests. We report a UK-based economic evaluation of these realistic strategies to identify individuals with monogenic diabetes (defined here as mutations in *GCK, HNF1A* or *HNF4A* genes). The development of the model-based economic evaluation has been published elsewhere.[12] The economic evaluation was undertaken alongside a clinical study whose aims included (1) Investigating the prevalence of monogenic diabetes within two areas of the UK. (2) Measuring the effects of a change of treatment following a positive diagnosis of monogenic diabetes. The clinical study recruited 1407 individuals who were diagnosed with diabetes <30 years old and who were <50 years old at recruitment.[13] Prospective quality of life using the EuroQol (EQ-5D) Index, a generic measure of health outcome[14] and glycated haemoglobin (HbA1c) data for 45 individuals who were diagnosed with monogenic diabetes within the geographical areas of the clinical study were collected until 12 months after the genetic test result. Although the clinical study collected data on clinical outcomes, it was not designed, nor powered, to detect small changes in clinical outcomes. No statistically significant change in the EQ-5D Index or HbA1c before and 12 months after changing treatment was observed making it impossible to confirm or refute the clinically suspected benefit of changing treatment in persons found to have monogenic diabetes, but on inappropriate treatment. Thus, for the main analyses, only costs are considered in this economic evaluation, making this a conservative analysis of the testing strategies if patient benefit does occur. The implications of this are considered in the discussion.

The aim of this analysis is to evaluate and compare the lifetime costs of different realistic strategies in the NHS to identify individuals with monogenic diabetes and change their treatment to more appropriate therapy. This economic evaluation has been reported in line with the Consolidated Health Economic Evaluation Reporting Standards.[15]

## MATERIALS AND METHODS
### Model overview
A hybrid decision model was developed from the perspective of the NHS in England and Wales. A decision tree was developed in MicroSoft Excel to estimate the short-term (16 months) costs, which allowed a maximum of 4 months from referral to testing to change of treatment (for those identified as having monogenic diabetes), plus 12 months follow-up (coinciding with the accompanying clinical study). The IMS Centre for Outcomes Research (CORE) Diabetes Model (IMS CDM) V.8.5[16] was used to estimate the lifetime costs associated with the strategies. Expert consultation and explicit critical appraisal of existing long-term diabetes models helped to inform the structure of the decision model and choice of the IMS CDM (see Peters *et al*[12] for more detail on model

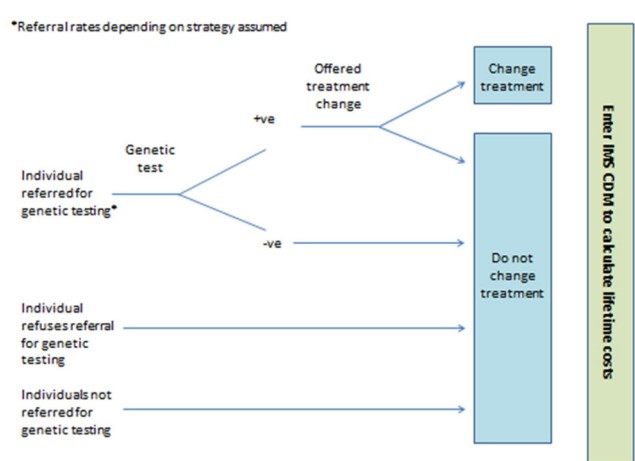

**Figure 1** Simplified model structure for the Ad Hoc Testing, Clinical Prediction Model Testing and All Testing strategies. IMS CDM, IMS CORE Diabetes Model.

development). Evidence to inform the model came from a number of sources including published and unpublished data and clinical opinion. Details on the evidence used in the model are given below.

### Strategies and comparator
Five strategies for identifying monogenic diabetes in individuals who were diagnosed with diabetes under the age of 30 years were defined: no genetic testing ('No Testing'), clinician-based genetic test referral ('Ad Hoc Testing'), targeted genetic testing based on clinical prediction models[9] ('Clinical Prediction Model Testing') or biochemical (urinary C peptide to creatinine ratio, UCPCR)[10] and immunological (islet autoantibodies)[11] test results ('Biomarker Testing'), blanket genetic testing ('All Testing').

The No Testing strategy is the comparator for all other strategies, as it represents the current policy within England and Wales where there is no guidance on the identification of individuals with monogenic diabetes. Thus, in this strategy all individuals remain on the diabetes treatment they were receiving at the start of the model, regardless of whether they truly have monogenic diabetes or not.

The Ad Hoc Testing strategy assumes no systematic referral of individuals for monogenic diabetes genetic testing. Instead, individuals are referred on an ad hoc basis depending on the awareness of local clinicians of monogenic diabetes (see figure 1). Data on referral rates for monogenic diabetes genetic testing in the UK[7] were used to calculate estimates of sensitivity and specificity of ad hoc referral.

In the Clinical Prediction Model Testing strategy, it is assumed that an individual general practitioner (GP) would complete the online monogenic diabetes prediction model (http://www.diabetesgenes.org/content/mody-probability-calculator)[9] to calculate a probability of the individual having monogenic diabetes (see figure 1). Depending on the probability of the individual having monogenic diabetes as calculated from the prediction

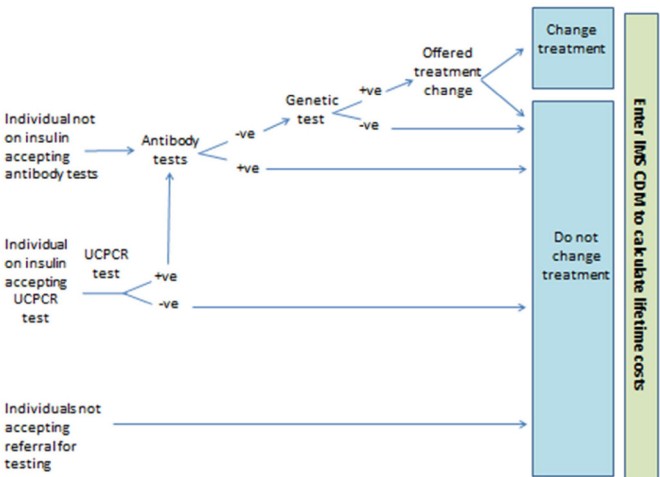

**Figure 2** Simplified model structure for the Biomarker Testing strategy. IMS CDM, IMS CORE Diabetes Model; UCPCR, urinary C peptide to creatinine ratio.

model, the GP would then refer them for monogenic diabetes genetic testing or not. Two versions of the prediction model exist, one to distinguish type 1 diabetes from monogenic diabetes (version 1) and the other to distinguish type 2 diabetes from monogenic diabetes (version 2). If the individual is currently receiving insulin, then version 1 of the prediction model is used, otherwise version 2 is used. For each version of the prediction model, nine thresholds are simulated in the decision model. Thus, the Clinical Prediction Model Testing strategy can be evaluated at 81 thresholds (9 from version 1 × 9 from version 2) for the simulated population. The decision model can then be used to identify the probability threshold for the prediction model that maximises the costs saved using the Clinical Prediction Model Testing strategy compared with the No Testing strategy.

In the Biomarker Testing strategy individuals receive biochemical and/or immunological tests depending on their demonstrated ability to produce insulin (see figure 2). If individuals are currently receiving insulin treatment, they are offered a UCPCR test to determine whether they are producing insulin or not.[10] Those with a positive UCPCR test are then offered a test for glutamic acid decarboxylase (GAD) and islet antigen 2 (IA2) autoantibodies.[11] If individuals are not currently receiving insulin treatment it is assumed they can produce their own insulin and so do not require a UCPCR test. Instead, those individuals not on insulin treatment are offered a test for GAD and IA2 autoantibodies. The aim of the GAD and IA2 autoantibodies test is to rule out those individuals with type 1 diabetes who are still producing insulin (ie, in the 'honeymoon' period). Individuals not showing the presence of autoantibodies are then offered the monogenic diabetes genetic test. In the All Testing strategy, all individuals are offered monogenic diabetes genetic testing (see figure 1).

## Model input parameters

### Population characteristics

The main analysis (modelled Cohort 1) simulated a prevalent cohort of individuals in England and Wales who were diagnosed with diabetes when <30 years old and were <50 years old at the start of the model. The prevalence of monogenic diabetes assumed in this cohort is 2.4% (GCK mutation 0.7%, HNF1A mutation 1.5%, HNF4A mutation 0.2%). A subgroup analysis (modelled Cohort 2) was undertaken to represent a future incident cohort who would have had a diagnosis of diabetes for a shorter duration than those in Cohort 1. Cohort 2 is defined as individuals diagnosed with diabetes when <30 years old and who were <30 years old at the start of the model, leading to a prevalence of 2.2% having monogenic diabetes. All information relevant to Cohort 2, including parameter values and results, are in online supplementary data 1. Further data on the prevalence and characteristics of Cohort 1 are given in online supplementary data 2.

### Test characteristics

Details of the test sensitivity and specificity used in the model are shown in online supplementary data 3. To calculate the sensitivity and specificity of referral for monogenic diabetes genetic testing in the Ad Hoc Testing strategy, four data sets were used:

▶ Diabetes prevalence from unpublished data for Tayside, Scotland.
▶ Estimates of total population by age and area from national census.[17]
▶ Monogenic diabetes prevalence from the accompanying clinical study.[13]
▶ Monogenic diabetes genetic test referral rates.[7]

The referral rates for monogenic diabetes genetic testing varied across the UK, with higher referral rates in areas where there is a strong research interest in monogenic diabetes, for example, the south-west of England, and Scotland. Estimates of sensitivity and specificity varied from sensitivity of 0.038 and specificity of 0.996 (Northern Ireland) to sensitivity 0.196 and specificity 0.977 (south-west of England) (see online supplementary data 3). To account for the general low rates of referral in the UK, we assumed the referral rates for one of the lowest areas, northern Ireland. In sensitivity analyses, data from all individual regions were used to estimate sensitivity and specificity for the Ad Hoc Testing strategy. However, the cost of increased awareness in one area compared with other areas is not known, and so it is not possible to estimate the additional cost of increased awareness of monogenic diabetes in the Ad Hoc Testing strategy, such as the south-west of England and Scotland.

For the Clinical Prediction Model Testing strategy the probability thresholds of 10%–90% for the two versions of the test were taken from Shields et al,[9] with sensitivity ranging from 0.5 to 0.99 and specificity ranging from 0.65 to 0.996. All 81 combinations of probability thresholds were evaluated in the decision model. No adjustments were made to the clinical prediction model as the

population on which it would be applied (individuals with diabetes in England and Wales) is very similar to that on which it is based. In the Biomarker Testing strategy, sensitivity of 0.94 and specificity of 0.96 for the UCPCR test was used based on a UCPCR cut-off of ≥0.2 nmol/mmol to discriminate individuals with *HNF1A* and *HNF4A* mutations who were insulin-treated from individuals with type 1 diabetes.[10] Besser *et al* did not report on the sensitivity and specificity of this cut-off to discriminate insulin-treated type 2 from *GCK*, *HNF1A* and *HNF4A* mutations, or to discriminate type 1 from *GCK* mutations. Since use of a different UCPCR cut-off for type 1 or insulin-treated type 2 would be difficult in practice (Besser *et al*[10]), we assumed that the UCPCR cut-off of ≥0·2 nmol/mmol could be used to discriminate type 1 from insulin-treated type 2, *HNF1A* and *HNF4A* mutations. Furthermore, Besser *et al* report that UCPCR cannot be used to discriminate *GCK* from *HNF1A* and *HNF4A* mutations. Thus, we assume that the UCPCR cut-off of ≥0.2 nmol/mmol can be used to discriminate type 1 diabetes from insulin-treated type 2, *GCK*, *HNF1A* and mutations. The impact on the model results of using different estimates of sensitivity and specificity is assessed in sensitivity analyses. Data from McDonald *et al*[11] were used to inform the sensitivity and specificity for the GAD and IA2 autoantibody tests (see online supplementary data 3). For all testing strategies, individuals referred for the monogenic diabetes genetic test were either tested for mutations in the *GCK* gene only, the *HNF1A* and *HNF4A* genes together, or all three genes (see online supplementary data 2).

### Uptake and repeat tests

Using data from the accompanying clinical study, for Cohort 1, it was assumed that 8·2% of individuals would decline the offer of genetic testing (6.9% for Cohort 2). This percentage was applied to all of the strategies where genetic testing was an option. For the Biomarker Testing strategy it was assumed that 11·9% of Cohort 1 (12.8% of Cohort 2) individuals offered the UCPCR test and 8·2% of Cohort 1 (6.9% of Cohort 2) individuals offered the autoantibody test would not accept. Estimates of the number of repeat tests required for both cohorts in the Biomarker Testing strategy are reported in online supplementary data 2.

### Family genetic testing

It was assumed in the model that identification of an individual with monogenic diabetes from any of the defined strategies would lead to first degree family members (who fit the defined cohort) also being genetically tested. Once individuals identified from the testing strategies have had the genetic test and are found to have monogenic diabetes, their family members receive the monogenic diabetes genetic tests. In Cohort 1, it was assumed that for every 10 individuals identified by the testing strategies as having monogenic diabetes, a further 6·3 family members are genetically tested, with 5.9 of these assumed to have the mutation (based on UK referral rate data).[7] These

ratios were applied to the Ad Hoc Testing, Clinical Prediction Model Testing and Biomarker Testing strategies.

### Treatment for diabetes

The treatment pattern assumed at the model start is given in online supplementary data 2. These data are from the accompanying clinical study where the treatment pattern for those truly having monogenic diabetes is based on just 45 individuals. The impact on the model results of the type of treatment at the start of the model is assessed in sensitivity analyses. Only individuals with a positive genetic test were offered a treatment change, which was cessation of diabetes treatment for those with the *GCK* mutation or to sulphonylureas for individuals with the *HNF1A* or *HNF4A* mutations. Data from the clinical study informed the likely treatment pattern once individuals are diagnosed with monogenic diabetes. For Cohort 1, at 1 month after treatment change it was assumed that 86% of individuals with *HNF1A* or *HNF4A* mutations were receiving a more appropriate treatment, at 3 months this was 86%, at 6 months this was 89% and at 12 months this was 77% (see online supplementary data 2). Some individuals having a positive genetic test result may not successfully change to sulphonylurea treatment alone and may continue to receive insulin.[18] For individuals with *HNF1A* or *HNF4A* mutations it was assumed that they would require insulin treatment eventually, and how much insulin and when they would start taking it would depend on whether they had previously received sulphonylureas and progressed to insulin or had started on insulin initially. As no data are available, two experts in monogenic diabetes (AH and EP) were consulted for their opinion (see online supplementary data 2). Based on data from the accompanying clinical study it was assumed that 93% of individuals identified to have the GCK mutation, would successfully stop all diabetes treatment.

### Resource use

The type of NHS costs (£), inflated to 2018 prices using the Hospital and Community Health Services pay and prices index[19] considered within each strategy are summarised in (online supplementary data 4).

All treatment costs were estimated using the reported doses from the clinical study and the British National Formulary.[20] The costs associated with the tests include costs for the collection of blood and urine samples, costs of the UCPCR and autoantibody tests, and genetic test costs. The costs of nurse time spent providing assistance to those individuals with monogenic diabetes who are changing to a more appropriate treatment were also included (see online supplementary data 4) .

The costs associated with home blood glucose monitoring (HBGM) were also included in the model. The frequency of HBGM before and after diagnosis of monogenic diabetes, and any subsequent change in treatment, were estimated from the clinical study for individuals truly having monogenic diabetes (see online supplementary data 2). Data from the literature were used to inform

HBGM frequency in individuals with type 1 and type 2 diabetes.[21] [22] It was assumed that individuals who have a *GCK*, *HNF1A* or *HNF4A* mutation, but did not have a genetic test or change treatment would have the same HBGM frequency as at the start of the model. Costs of HBGM were based on use of the Accu-Check Aviva metre (£16.09 for 50 strips).[20]

The costs of diabetes-related complications for individuals with type 1 diabetes, type 2 diabetes, and *HNF1A* or *HNF4A* mutations were identified from reviewing the published literature and using data from the National Schedule of Reference Costs 2016/2017. Only cost data from the UK were modelled in the IMS CDM (see online supplementary data 4). The majority of cost estimates from the literature were associated with uncertainty, mainly in inflating the costs to 2018 due to the age of the evidence available, therefore all of the long-term costs inputted into the model were rounded to the nearest £50 to avoid spurious precision. It is assumed that individuals with *GCK* mutations do not experience long-term diabetes-related complications[3] and once identified as having a mutation in the GCK gene, they no longer incur the costs of diabetes-specific consultations. Data from Curtis 2017[19] and Currie *et al* 2010[23] were used to inform the costs of diabetes-specific consultations (see online supplementary data 4).

### Long-term events and survival

It was assumed that individuals with *GCK* mutations do not experience diabetes-related events and have the same mortality rate as the general population.[17] Therefore individuals with GCK mutations do not enter the IMS CDM. For individuals with *HNF1A* and *HNF4A* mutations, due to limited data on long-term complications and mortality, it was assumed that these individuals have the same pattern of long-term complications and mortality as individuals with type 1 diabetes. Therefore individuals with *HNF1A* and *HNF4A* mutations were modelled using the type 1 diabetes model in the IMS CDM.

### Model outcomes

All costs (£, 2018) beyond the first year are discounted at a rate of 3·5% per annum to account for the preference for deferring future costs in economic evaluations.[24] Discounted and undiscounted total costs are reported in the results section alongside the estimated discounted incremental costs per person with diabetes over a lifetime for each strategy compared with the No Testing strategy and the proportion of monogenic diabetes cases identified by each strategy.

### Analysis

The results of a 'base case' analysis are presented, but due to the uncertainty surrounding many of the parameter estimates, alternative combinations of assumptions may be equally plausible. Therefore, wide-ranging one-way sensitivity and threshold analyses have been conducted to explore the different sources of uncertainty; this includes an analysis where an improvement in utility for those who successfully change treatment is assumed. Details of the sensitivity and threshold analyses undertaken for Cohort 1 can be found in online supplementary data 2 (see online supplementary data 1 for details on Cohort 2 analyses). In contrast to our planned analysis,[12] we decided not to do a probabilistic analysis because important structural uncertainties in this model could not be fully captured by a probabilistic analysis (it would therefore be misleading).

### Patient and public involvement

There was no patient and public involvement in the development or analysis of the model.

### RESULTS
### Cohort 1: diagnosed <30 years old, <50 years old at start of model

For the 'base case' analysis, the total discounted costs per person with diabetes over a lifetime were estimated to be £53 600–£54 000 depending on the strategy used (see table 1). The All Testing strategy was estimated as

**Table 1** Summary of the per person lifetime costs* and percentage of cases and non-cases genetically tested for each strategy (ordered by increasing cost of strategy)

| Strategy | Total undiscounted costs* | Total discounted costs* | Incremental costs versus no testing strategy* | % who are genetically tested | |
|---|---|---|---|---|---|
| | | | | With monogenic diabetes | Without monogenic diabetes |
| Clinical Prediction Model Testing† | £133 200 | £53 600 | −£100 | 92 | 3 |
| Biomarker Testing | £133 300 | £53 600 | −£100 | 92 | 8 |
| Ad Hoc Testing | £133 500 | £53 700 | 0 | 6 | <1 |
| No Testing | £133 600 | £53 700 | NA | 0 | 0 |
| All Testing | £133 700 | £54 000 | £300 | 92 | 92 |

*Rounded to nearest £100.
†Probability thresholds chosen to maximise costs saved versus No Testing are 12.6% for type 1 versus monogenic diabetes and 75.5% for type 2 versus monogenic diabetes.

the most costly (£54,000), the cheapest options were the Clinical Prediction Model Testing (where the probability thresholds were chosen to maximise costs saved compared with No Testing) and Biomarker Testing strategies (£53,600). The No Testing and Ad Hoc Testing strategies were both estimated as £53 700 per person with diabetes over a lifetime. The Ad Hoc Testing strategy was estimated to identify very few cases of monogenic diabetes (6%) compared with the All Testing strategy which was estimated to identify 92% of monogenic diabetes cases. No more than 92% of monogenic diabetes cases can be identified by any strategy due to the assumption that 8% of individuals will not accept an offer of genetic testing for monogenic diabetes. Family testing boosts the detection of monogenic diabetes cases to 92% in the Clinical Prediction Model Testing and Biomarker Testing strategies. The costs saved for these two strategies over the No Testing strategy relate to more individuals getting a monogenic diabetes diagnosis and changing to receive more appropriate treatment which is cheaper and also leads to a reduction in the frequency of HBGM. The All Testing strategy is the most expensive since although more monogenic diabetes diagnoses are made, resulting in fewer treatment and HBGM costs, the costs of genetically testing all individuals diagnosed with diabetes are very high.

As there are 81 different combinations of probability thresholds for the clinical prediction model, the combination of thresholds which maximises the costs saved for the Clinical Prediction Model Testing strategy have been reported above. In figure 3, all 81 threshold combinations for the clinical prediction model are shown. The Clinical Prediction Model Testing strategy is estimated to identify 74%–92% of monogenic diabetes cases depending on the probability threshold combinations used to refer individuals for genetic testing. The lifetime costs saved per person with these threshold combinations compared with No Testing vary from £0 to £150.

Sensitivity analysis results suggest that the impacts on costs in the different scenarios are insensitive to wide-ranging, plausible changes to key model parameters, (see

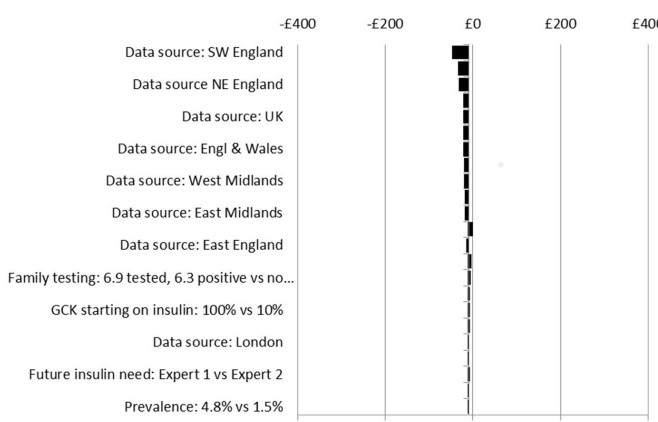

**Figure 4** Sensitivity analyses: incremental costs per person over a lifetime for the Ad Hoc Testing strategy versus the No Testing strategy. *GCK*, glucokinase.

figures 4–7). No plausible parameter value changes the finding that the Ad Hoc Testing and Clinical Prediction Model Testing strategies are always estimated to save costs compared with the No Testing strategy. Only extreme assumptions on the uptake of genetic and UCPCR testing (just 10% uptake) suggest fewer costs are saved from the Biomarker Testing strategy when compared with the No Testing strategy. Except for assumptions on test uptake, the estimated cost savings are in the region of £0–£50 per person over a lifetime for the Ad Hoc Testing strategy (see figure 4), £50–£300 for the Clinical Prediction Model Testing strategy (see figure 5) and £50–£250 for the Biomarker Testing strategy (see figure 6). The All Testing strategy is estimated to cost an additional £150–£350 per person over a lifetime compared with the No Testing strategy except when the cost of the genetic test is assumed to be <60% of its current cost (see figure 7).

As figures 4–7 show, the findings are most sensitive to:
► The estimated prevalence of monogenic diabetes within the cohort—increasing prevalence (from 2.4% to 4.8%) leads to greater costs saved for the Ad

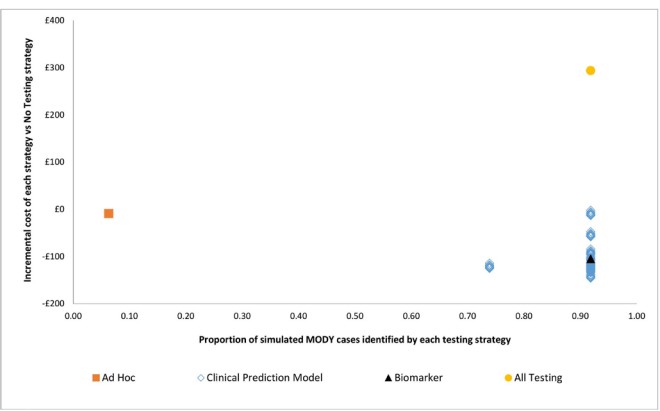

**Figure 3** Base case incremental costs (vs No Testing) and the proportion of monogenic diabetes cases identified for each strategy.

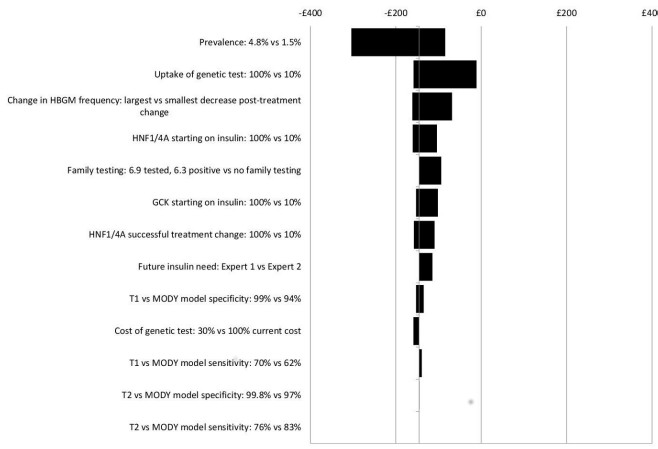

**Figure 5** Sensitivity analyses: incremental costs per person over a lifetime for the Clinical Prediction Model Testing strategy versus the No Testing strategy. *GCK*, glucokinase; HBGM, home blood glucose monitoring; HNF1/4A, hepatocyte nuclear factor 1/4 alpha.

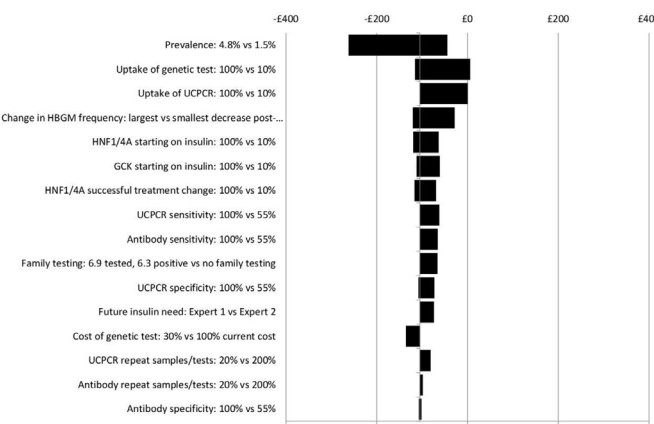

**Figure 6** Sensitivity analyses: incremental costs per person over a lifetime for the Biomarker Testing strategy vs the No Testing strategy. *GCK*, glucokinase; HBGM, home blood glucose monitoring; HNF1/4A, hepatocyte nuclear factor 1/4 alpha; UCPCR, urinary C peptide to creatinine ratio.

Hoc Testing, Clinical Prediction Model Testing and Biomarker Testing strategies compared with the No Testing strategy.

► The uptake of testing—reduced uptake leads to fewer costs saved for all strategies compared with the No Testing strategy,

► The frequency of HBGM pretreatment and post-treatment change—assuming that individuals change their frequency of HBGM by only a small amount after a diagnosis of monogenic diabetes, leads to fewer costs saved compared with the No Testing strategy,

► The proportion of individuals with monogenic diabetes who receive insulin before their monogenic diabetes diagnosis—the larger the proportion receiving insulin before being diagnosed as having monogenic diabetes, the greater the costs saved for all strategies compared with No Testing.

Threshold analysis results (see online supplementary data 2) suggest that when the genetic tests are reduced to

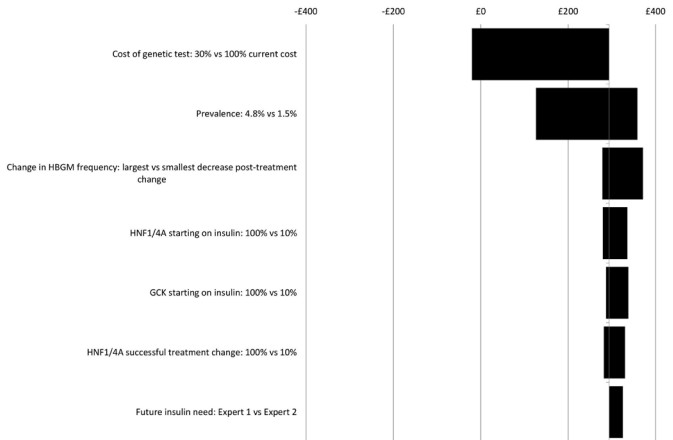

**Figure 7** Sensitivity analyses: incremental costs per person over a lifetime for the All Testing strategy versus the No Testing strategy. *GCK*, glucokinase; HBGM, home blood glucose monitoring; HNF1/4A, hepatocyte nuclear factor 1/4 alpha.

approximately 35% of their current costs, the All Testing strategy incurs no additional costs compared with the No Testing strategy. However, in this situation, the Biomarker Testing and Clinical Prediction Model Testing strategies are estimated to save, approximately £150 per person over a lifetime, compared with the No Testing strategy. Reducing the percentage of individuals with monogenic diabetes who are receiving only insulin at the start of the model has little impact on the incremental costs estimated: even if 10% of individuals with *GCK* mutations or 10% of individuals with *HNF1A* or *HNF4A* mutations are on tablets at the start of the model, slight cost savings are still estimated with the Clinical Prediction Model Testing and Biomarker Testing strategies compared with the No Testing strategy (see figures 5 and 6).

Threshold analyses specific to the Biomarker Testing strategy demonstrate that once uptake of the UCPCR and autoantibody tests is reduced to less than 70%, the costs saved with the Biomarker Testing strategy compared with the No Testing strategy reduce. Costs saved with the Biomarker Testing strategy are most sensitive to reductions in the sensitivity of the UCPCR and autoantibody tests. Increases in the number of repeat urine or blood samples and tests required within the Biomarker Testing strategy have little impact on the estimate of costs saved compared with the No Testing strategy.

### Cohort 2: diagnosed <30 years, <30 years at start of model

As in Cohort 1, the Clinical Prediction Model Testing and Biomarker Testing strategies are estimated to save £100 per person with diabetes over a lifetime compared with the No Testing strategy, while the All Testing strategy is assumed to cost an additional £300 compared with the No Testing strategy. When compared with Cohort 1, the Clinical Prediction Model Testing and Biomarker Testing strategies are not estimated to save any more costs because of the trade-off between individuals being less likely to be on insulin prior to genetic testing in Cohort 2 (67% vs 83% in Cohort 1) even though they are more likely to successfully change to sulphonylureas than Cohort 1 (100% vs 79% in Cohort 1). Individuals in Cohort 2 were estimated to monitor their blood glucose less frequently before receiving a diagnosis of monogenic diabetes compared with Cohort 1, and so fewer costs are saved from reducing further the HBGM frequency than is the case for Cohort 1. See online supplementary data 1 for further results, including sensitivity analyses which suggest that estimates of prevalence and testing uptake have the largest impact on the findings (as for Cohort 1).

### DISCUSSION

The Clinical Prediction Model Testing and Biomarker Testing strategies modelled here have been estimated to be cost saving for identifying individuals with monogenic diabetes and changing their treatment compared with the current practice of no genetic testing. Assumptions about the prevalence of monogenic diabetes within the

simulated cohort, the uptake of testing and the frequency of HBGM before and after receiving a diagnosis of monogenic diabetes had the largest impact on the findings, but did not change the overall conclusions that targeted strategies are estimated to save costs compared with the No Testing or All Testing strategies. Data on prevalence and test uptake were taken directly from the accompanying clinical study, which is the first to systematically estimate prevalence of monogenic diabetes in the UK.[13] Information on the frequency of HBGM before and after a diagnosis of monogenic diabetes is based on just a small number of individuals, but is currently the best evidence available.

This is the first UK-based economic evaluation of strategies to identify individuals with monogenic diabetes. A published paper documented the development of the model and the intended analysis,[12] and the minor departures from the protocol have been declared and justified. UK data have been used to inform many of the model inputs for which there was previously no credible evidence. However, due to the rarity of monogenic diabetes, many inputs specific to individuals with monogenic diabetes are based on very few individuals, especially for Cohort 2, or assumptions. For instance, it was assumed that treatment and HBGM frequency data taken from the clinical study at 12 months follow-up remained constant over time in the model, with additional long-term treatment data informed by clinical opinion. Until longer follow-up data are available, it is unclear what impact these assumptions may have on the model results.

We simulated two cohorts, both based on data from the clinical study. The aim of Cohort 2 was to assess the impact of strategies for identifying monogenic diabetes in individuals more recently diagnosed with diabetes than those in Cohort 1. Although it was anticipated that individuals in Cohort 2 would find it easier to change to more appropriate treatment (because they had not been on their existing treatment for a long time), we actually found that individuals in Cohort 2 were less likely to be on insulin at that point, so costs saved from changing treatment were smaller than for Cohort 1, even though more individuals changed treatment. However this analysis was limited by the low number of participants close to diagnosis for which data were available. Furthermore, the performance of the Clinical Prediction Model Testing and Biomarker Testing strategies are based on prevalent cohorts[9–11] which will impact on their generalisability to an incident cohort (Cohort 2). Thus, there are still many uncertainties associated with the results, including that the IMS CDM has not been validated for monogenic diabetes, so these results should be interpreted with this in mind. Nevertheless, the numerous sensitivity and threshold analyses estimated cost savings for the Clinical Prediction Model Testing (when choice of thresholds was maximised to save costs) and Biomarker Testing strategies compared with No Testing.

Naylor *et al*[25] conducted an economic evaluation of genetic testing (akin to our All Testing strategy) for monogenic diabetes in individuals aged 25–40 years who were newly diagnosed with type 2 diabetes compared with no genetic testing from a US health system perspective. Individuals identified as having *HNF1A* or *HNF4A* mutations who successfully transferred to sulphonylureas were assumed a HbA1c reduction of 16.4 mmol/mol compared with those not changing treatment (based on six individuals at 3 months follow-up after treatment change)[26] and a utility increase of 0·13 for transferring from insulin to sulphonylurea treatment (based on evidence from 519 individuals aged 65 years and older with type 2 diabetes).[27] Naylor *et al* reported a gain of 0·012 quality-adjusted life-years (QALYs) for the testing strategy at an additional cost of $2400 per person over a lifetime compared with their no testing strategy, resulting in an incremental cost-effectiveness ratio of $205 000 per QALY gained.[25] The additional costs for the genetic testing strategy in Naylor *et al*[25] are much greater than the All Testing strategy in our evaluation ($2400 vs £300) because of differences in the populations simulated. In our evaluation a younger diabetes population is assumed, with individuals who truly have monogenic diabetes being more likely to be misdiagnosed with type 1 and receive insulin. The simulated population in Naylor *et al* is older and explicitly those diagnosed with type 2, therefore are less likely to receive insulin treatment, so have fewer cost savings from changing treatment.

The health impacts assumed by Naylor *et al*[25] are also different from those observed in our accompanying clinical study. Using the EQ-5D Index, we found little evidence over the 12 months treatment change period for an improvement in utility associated with more appropriate treatment, although the EQ-5D visual analogue scale and the Diabetes Treatment Satisfaction Questionnaire did suggest an improvement at 12 months. Furthermore, in the sample of 28 individuals with *HNF1A* or *HNF4A* mutations who successfully changed to sulphonylureas, no statistically significant impact on HbA1c at 12 months after treatment change was found (mean difference of 3·43 mmol/mol (95% CI −2·18 to 9·04)). Due to the lack of evidence suggesting an effect on quality of life and HbA1c we took the decision to assume there were no differences in quality of life and HbA1c between those identified as having monogenic diabetes and subsequently changing treatment, and those not identified. Our evaluation was conservative, as evidence shows that changing treatment can have a substantial beneficial impact on individuals.[28 29] A sensitivity analysis assuming an improvement in utility for those found to have *HNF1A* or *HNF4A* mutations who successfully changed treatment indicated <5 quality-adjusted days were gained from the Clinical Prediction Model, Biomarker and All Testing strategies compared with No Testing. However, generic and relatively simple quality of life measures (eg, EQ-5D) are likely to be insensitive to the magnitude and type of changes individuals with diabetes might experience when changing to more appropriate treatment. Measuring such changes to quality of life is also limited by the ceiling

effect, since these individuals generally constitute a well-controlled, young diabetes population with a good quality of life. Given these limitations we have not considered any reductions in quality of life that may occur during the testing period, especially for those tested but not found to have monogenic diabetes.

A further limitation is in the evidence used to inform the sensitivity and specificity of the testing strategies. For example, the accuracy of antibody testing for the Biomarker strategy is based on a two-gate study design where the test is evaluated by comparing test results in individuals known to have a diagnosis of monogenic diabetes with those newly diagnosed with type 1 diabetes. Such study designs have been shown to lead to overstated accuracy estimates.[30]

A limitation of the Ad Hoc Testing strategy is in choosing the referral rates that are representative. We used referral rates for the area with the lowest rate of referral. We could have used an average referral rate across the country, but would not have been able to capture the relevant costs of the increased awareness in some areas (such as the south-west of the UK where the referral centre for monogenic diabetes is based) which is linked to increased referral.

The results suggest that within the context of the NHS, the additional costs of genetically testing (a relatively large number of) individuals are likely to be offset by the lifetime savings from the subsequent treatment changes in a very small proportion of individuals. Although the estimated cost savings are relatively small per person (approximately £100–£200 over a lifetime), assuming there are approximately 200 000 individuals (personal communication) in England and Wales who are <50 years old and have had a diagnosis of diabetes before the age of 30 years, between £20 million and £40 million could be saved if such strategies are used. To be able to apply these findings to other populations the cost of the testing in particular will need to be updated. If the genetic test costs are significantly higher, then it is unclear whether the Clinical Prediction Model Testing and Biomarker Testing strategies could be considered cost saving, or even cost neutral. However, further collection of treatment pattern, HBGM frequency, HbA1c and quality of life data for individuals with monogenic diabetes is required to better inform the decision model, especially to model an incident cohort. Additional strategies to better identify those with monogenic diabetes are feasible, and in development, but will also require evaluation for their effectiveness and cost-effectiveness.

## CONCLUSIONS

Targeted strategies to identify individuals with monogenic diabetes and change to more appropriate treatment may be cost saving to the NHS. However, collection of longer-term treatment and frequency of HBGM data would be valuable to reduce the main uncertainties in the modelling. Future work to evaluate the use of genetic testing strategies soon after diagnosis of diabetes would be useful to policy makers.

**Acknowledgements** The authors thank IMS Health for use of the IMS CDM.

**Contributors** JLP designed the decision model, contributed to data collection, undertook analysis and interpretation of the model results and drafted the manuscript. RA and CH helped design and analyse the decision model, and contributed to the interpretation of the results and drafting of the manuscript. BS, MH, MS, TJM, EP and AH contributed to the study design and data collection, and commented on the manuscript. SK contributed to data collection and commented on the manuscript.

**Funding** This study was supported by the Department of Health and Wellcome Trust Health Innovation Challenge Award (HICF-1009-041 and WT-091985). JLP is partly supported by the NIHR Collaboration for Leadership in Applied Health Research and Care for the South West Peninsula (PenCLAHRC). BS, MH and AH are core members of the National Institute for Health Research (NIHR) Exeter Clinical Research Facility. TJM is supported by an NIHR Chief Scientist Office Fellowship. AH is a Wellcome Trust Senior Investigator and an NIHR Senior Investigator. EP is a Wellcome Trust New Investigator. MS is a National Institute for Health Research (NIHR) Senior Nurse and Midwife Research Leader (NIHR4-SNMRL058) and is also supported by the Exeter NIHR Clinical Research Facility at the University of Exeter.

**Competing interests** None declared.

**Patient consent for publication** Not required.

**Provenance and peer review** Not commissioned; externally peer reviewed.

**Data availability statement** No data are available. The decision analytic model described in this manuscript is not available due to the IMS CDM being under license for the current study.

**Author note** Sophie King is currently at Centre for Trials Research, Cardiff University, Cardiff, UK

**ORCID iDs**
Jaime L Peters http://orcid.org/0000-0003-1778-3518
Michelle Hudson http://orcid.org/0000-0003-0623-1873
Maggie Shepherd http://orcid.org/0000-0003-2660-0955

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
