## [Reviewer comments · BMJ Open]

ARTICLE DETAILS

TITLE (PROVISIONAL)	Strategies to Identify Individuals with Monogenic Diabetes: Results of an Economic Evaluation
AUTHORS	Peters, Jaime; Anderson, Rob; Shields, Beverley; King, Sophie; Hudson, Michelle; Shepherd, Maggie; McDonald, Timothy; Pearson, Ewan; Hattersley, Andrew; Hyde, Chris

VERSION 1 – REVIEW

REVIEWER	Rochelle Naylor University of Chicago United States
REVIEW RETURNED	08-Nov-2019

GENERAL COMMENTS	Peters et al., have completed an elegant and important study to add support to wider implementation of genetic testing and precision medicine in monogenic diabetes. While two recent CEAs (Johnson et al. and Goodsmith et al, of which I am co-author) have shown cost-savings in genetic testing strategies for monogenic diabetes, both were restricted to pediatric populations. This is the first study carried out in the UK which has substantial experience with and widest capture of patients with monogenic diabetes. It is the first study to show cost-effectiveness in a population that includes adults. A significant additional strength of this study is much of the data were modeled on a recent study that was intentionally conducted in parallel with this study. The study was rigorously performed, according to appropriate practices in disease modeling and CEA. Limitations of the study are those expected in disease models, particularly of an uncommon disease, and CEA- namely, data from small cohorts and model parameter uncertainty. The authors have conducted appropriate sensitivity analyses based on plausible ranges of various model inputs. Hopefully, this study will move the needle, not only in the UK, but broadly for timely access to genetic testing covered by health insurance. I did note some minor errors or needed clarifications: Page 6, Line 28- Delete "In the event" at the start of the sentence. Page 10, Line 35- Add Scotland after Tayside, as readers will be broadly located and not necessarily familiar with the region. Page 14, line 9- "an" should be and Table 1H. Second row, last column, line 12. Point 1 should state "all of the remaining..." In Table 1H and 2H, there appears to be some broken reference links, with these words appearing: "Error! Reference source not found."
--

REVIEWER	Aaron Winn Medical College of Wisconsin, USA
REVIEW RETURNED	25-Nov-2019

GENERAL COMMENTS	Overall, this is an excellent paper that answers a policy relevant question. The additional information in the appendix was very helpful to answer many questions that the reader may have. I have a few comments:  1. Clarity Around T1 or T2 Natural History Models: I had a hard time following what natural history model (T1 or T2) applied to each genetic mutation. It seems like there was a decision rule around using insulin (strategies and comparators section) but in the survival section it sounds like everything will be done with a T1 model. 2. GCK population: Does this group enter into the long-term diabetes simulation model since they do not experience diabetes related events? 3. Please provide why doing a probabilistic sensitivity analysis would be misleading - even if you can get a structural uncertainty related to disease progression, it would still be helpful for the reader to understand how varying the parameters you do have a good handle on would modify the result. 4. Maybe I missed it but a sensitivity analysis that incorporates quality of life changes would be interesting to reader if they are concerned that the EQ5D may not be sensitive enough to any quality of life changes that are unique to this clinical situation.
---

REVIEWER	Stephanie Johnson Queensland Children's Hospital, Australia
REVIEW RETURNED	27-Nov-2019

GENERAL COMMENTS	This study evaluates the costs associated with four different strategies for identifying monogenic diabetes. Compared to no-testing, the additional cost of testing was offset by the life-time savings for each of the testing strategies, aside from testing all participants and was cost-neutral for Ad hoc testing. This is an important consideration when recommending changes to clinical practice. Major points:  1. This study has only accounted for costs, not patient outcomes (preferably QALYs) in their model. This seems like a real missed opportunity given the efforts of required to develop such a model. Without understanding patient outcomes, information on costs is really somewhat meaningless. It doesn't provide any insight into the relative value of the different strategies, which is what is required for sensible decision making (i.e. to be able to maximise health benefits from a given pool of resources). 2. Previous studies from this group have demonstrated improvement in HbA1c and quality of life changing to sulphonylurea. The absence of any improvement need explanation. 3. Each of the four treatment strategies were compared to no testing. It may be more appropriate to use current testing rates as the comparator groups. The reason given for this was that there are no national (NHS) guidelines for identifying individuals with monogenic diabetes. However Shields et al 2010 demonstrated
--

	that although rates of testing varies throughout the UK, testing still occurred in each region. 4. The sensitivity for each of the testing strategies was based on the premise that <1% of the population with MODY had positive antibodies. This was based on a single study of rates of antibodies in 508 individuals referred for genetic testing, which in itself is a biased sample. The gold standard would be to sequence the entire diabetes population to establish actual sensitivity and specificities of each strategy. 5. The “Ad hoc” testing was based on the area with the lowest referral rates from a 2010 study. Presumably referral rates have increased with the increased awareness since then. Using an average referral rate for the UK may be more appropriate Minor points  1. It would be more logical to have cohort 1 as supplementary table 1 2. Overall there are too many figures:  a. Figure 1 and 2 could possibly be combined into one decision tree b. Figure 3 is unnecessary and could be included into the subsequent tornado plots c. Figure 4b would be more generalisable if identified prevalence was used rather than region-based data. There was no description as to how the rates were extrapolated from rates of MODY identified in each region from referrals for testing, population data and rates of diabetes in those <30ys etc 3. Supplementary table 1H states See Error! Reference source not found. 4. Supplementary Figures 1F-K and 2A-F are unnecessary and can be included in the tornado plots 5. Table 4C is not included in the analysis as there is no difference reported in outcomes between any of the groups. Thus costs of complications is irrelevant
--	--

VERSION 1 – AUTHOR RESPONSE

Reviewer 1

This is the first study carried out in the UK which has substantial experience with and widest capture of patients with monogenic diabetes. It is the first study to show cost-effectiveness in a population that includes adults. A significant additional strength of this study is much of the data were modeled on a recent study that was intentionally conducted in parallel with this study.

The study was rigorously performed, according to appropriate practices in disease modeling and CEA. Limitations of the study are those expected in disease models, particularly of an uncommon disease, and CEA- namely, data from small cohorts and model parameter uncertainty. The authors have conducted appropriate sensitivity analyses based on plausible ranges of various model inputs.

We thank and appreciate these comments from Reviewer 1.

Page 6, Line 28- Delete "In the event" at the start of the sentence.

Words deleted

Page 10, Line 35- Add Scotland after Tayside, as readers will be broadly located and not necessarily familiar with the region.

Done

Page 14, line 9- "an" should be and

Done

Table 1H. Second row, last column, line 12. Point 1 should state "all of the remaining..."
This has been amended.

In Table 1H and 2H, there appears to be some broken reference links, with these words appearing: "Error! Reference source not found."

These broken links have been removed.

Reviewer 2

1. Clarity Around T1 or T2 Natural History Models: I had a hard time following what natural history model (T1 or T2) applied to each genetic mutation. It seems like there was a decision rule around using insulin (strategies and comparators section) but in the survival section it sounds like everything will be done with a T1 model.

The decision rule (whether currently treated with insulin or no) relates to use of the clinical prediction model in the Clinical Prediction Model strategy. In terms of the natural history model, we assumed that individuals with the HNF1A and HNF4A mutations would have a pattern of complications and mortality more similar to individuals with type 1 diabetes than type 2 diabetes. Therefore we modelled all individuals with HNF1A and HNF4A mutations using the type 1 natural history model.

This has been clarified in the manuscript (page 15) by changing the subheading to "Long-term events and survival" and amending the text as follows:

Long-term events and survival

"It was assumed that individuals with GCK mutations do not experience diabetes related events and have the same mortality rate as the general population¹⁷. Therefore individuals with GCK mutations do not enter the IMS CDM. For individuals with HNF1A and HNF4A mutations, due to limited data on long-term complications and mortality, it was assumed that these individuals have the same pattern of long-term complications and mortality as individuals with type 1 diabetes. Therefore individuals with HNF1A and HNF4A mutations are modelled using the type 1 diabetes model in the IMS CDM."

2. GCK population: Does this group enter into the long-term diabetes simulation model since they do not experience diabetes related events?

As we assume that individuals with GCK mutations do not experience diabetes related events, they do not enter the simulation model. Instead it is assumed that they have the same mortality rate as the general population. This has been clarified in the manuscript (page 15) by changing the subheading to Long-term events and survival and amending the text as shown in the response above.

3. Please provide why doing a probabilistic sensitivity analysis would be misleading - even if you can get a structural uncertainty related to disease progression, it would still be helpful for the reader to understand how varying the parameters you do have a good handle on would modify the result.

We are concerned that spurious precision from conducting a probabilistic sensitivity analysis might be misleading, given the many uncertainties in the model that would not be captured in a probabilistic sensitivity analysis.

4. Maybe I missed it but a sensitivity analysis that incorporates quality of life changes would be interesting to reader if they are concerned that the EQ5D may not be sensitive enough to any quality of life changes that are unique to this clinical situation.

We discussed in the manuscript the difficulties in measuring any change in the EQ-5D utilities for the population of interest - the small sample size on which estimates are based, due to the rarity of monogenic diabetes; that many of the individuals in this population are young, with well-controlled diabetes so any improvements in utility cannot be estimated when most individuals are reporting 'perfect' utility at baseline.

However, we agree that a sensitivity analysis including the very small change in EQ-5D may be useful for readers. Therefore we have included a sensitivity analysis for Cohort 1 using the estimate of 0.02 improvement in EQ-5D utility 1 year after treatment change for the 14 individuals with HNF1A and HNF4A mutations who successfully changed treatment in the associated clinical study. We have stated in the manuscript that a sensitivity analysis on utility has been conducted (please see Analysis section, page 16). The methods and results of this sensitivity analysis has been added to the other sensitivity analyses in Supplementary Data 2. We have also referred to the results of this sensitivity analysis in the Discussion when comparing our findings to those from Naylor et al, please see page 24.

Reviewer 3

Major points:

1. This study has only accounted for costs, not patient outcomes (preferably QALYs) in their model. This seems like a real missed opportunity given the efforts of required to develop such a model. Without understanding patient outcomes, information on costs is really somewhat meaningless. It doesn't provide any insight into the relative value of the different strategies, which is what is required for sensible decision making (i.e. to be able to maximise health benefits from a given pool of resources).

In response to this comment and a similar comment from Reviewer 2, we have conducted a sensitivity analysis including the small change in utility observed in the individuals with HNF1A and HNF4A mutations who successfully changed treatment in the clinical study which was conducted alongside the modelling. Please see additional sensitivity analysis in Supplementary Data 2.

2. Previous studies from this group have demonstrated improvement in HbA1c and quality of life changing to sulphonylurea. The absence of any improvement need explanation.

The clinical study conducted alongside the modelling did identify improvements in quality of life and HbA1c. However these improvements were relatively small and not statistically significant. This may be due to many factors including the small sample size on which estimates are based, due to the rarity of MODY. A second factor, especially important for estimating utility, is that many of the individuals in this population are relatively young, with well-controlled diabetes. Thus, any improvements in utility cannot be estimated when most individuals are reporting 'perfect' utility at baseline.

Improvements in quality of life were found in EQ-5D VAS scores and on the treatment satisfaction questionnaire. We mention in the Discussion about the improvement found with the VAS scores, but

have now also added that improvements on the treatment satisfaction questionnaire were observed as follows:

“Using the EQ-5D Index, we found little evidence over the 12 month treatment change period for an improvement in utility associated with more appropriate treatment, although the EQ-5D visual analogue scale and the Diabetes Treatment Satisfaction Questionnaire did suggest an improvement at 12 months.” Please see page 24.

We choose to assume a conservative analysis by not including these small non-significant improvements in our model, however in response to comments we now provide a sensitivity analysis for the EQ-5D utility change. Please see Analysis section (page 16), Discussion section (page 24) and Supplementary Data 2 for details.

3. Each of the four treatment strategies were compared to no testing. It may be more appropriate to use current testing rates as the comparator groups. The reason given for this was that there are no national (NHS) guidelines for identifying individuals with monogenic diabetes. However Shields et al 2010 demonstrated that although rates of testing varies throughout the UK, testing still occurred in each region.

We had many discussions in our team about the most appropriate comparator for these analyses, and agree that there is evidence of testing in the UK, regardless of no national guidelines for this. The difficulty with comparing strategies to current practice (the Ad Hoc strategy defined in the manuscript), is that we have not been able to capture the costs associated with increased MODY awareness in different areas, thus the Ad Hoc strategy is estimated to be less costly than it will be in practice. Nonetheless if we compare strategies to the Ad Hoc strategies, the incremental costs do not change since the No Testing and Ad Hoc strategies have very similar costs (please see Table X below). Because of this and the reasons for originally choosing No Testing as the comparator, we believe remaining with No Testing as the comparator is a reasonable approach.

Table X

Strategy	Total discounted costs	Incremental costs compared to No Testing	Incremental costs to Ad Hoc Testing
Clinical Prediction Model	£53,600	-£100	-£100
Biomarker Testing	£53,600	-£100	-£100
Ad Hoc	£53,700	0	0
No Testing	£53,700	0	0
All Testing	£54,000	£300	£300

4. The sensitivity for each of the testing strategies was based on the premise that <1% of the population with MODY had positive antibodies. This was based on a single study of rates of antibodies in 508 individuals referred for genetic testing, which in itself is a biased sample. The gold standard would be to sequence the entire diabetes population to establish actual sensitivity and specificities of each strategy.

We agree that this is indeed a limitation of the performance of the strategies and have now highlighted this in the Discussion, please see page 25.

5. The “Ad hoc” testing was based on the area with the lowest referral rates from a 2010 study. Presumably referral rates have increased with the increased awareness since then. Using an average referral rate for the UK may be more appropriate

We agree that referral rates probably have increased since the time of the cited study. However, we are not able to estimate the costs associated with increased referral rates observed in some areas (for instance the costs associated with increased awareness of MODY in areas such as the South West and Scotland). To use an average referral rate would lead to improvements in the numbers being tested, but the costs associated with the improved awareness would not be accounted for.

In the original manuscript we mentioned this as a limitation of evaluating the Ad Hoc strategy, but have now also added this to the Discussion section, please see page 25.

Minor points

1. It would be more logical to have cohort 1 as supplementary table 1
The editorial style is to number Supplementary documents consecutively as they appear, therefore

2. Overall there are too many figures:
a. Figure 1 and 2 could possibly be combined into one decision tree
b. Figure 3 is unnecessary and could be included into the subsequent tornado plots
c. Figure 4b would be more generalisable if identified prevalence was used rather than region-based data. There was no description as to how the rates were extrapolated from rates of MODY identified in each region from referrals for testing, population data and rates of diabetes in those <30ys etc
We prefer to keep the figures as they are, but re-number Figures 4a-4d as indicated by the editor.

3. Supplementary table 1H states See Error! Reference source not found.
These broken links have been removed.

4. Supplementary Figures 1F-K and 2A-F are unnecessary and can be included in the tornado plots
We prefer to keep this figures as they are to show how the change in each parameter affects the incremental costs.

5. Table 4C is not included in the analysis as there is no difference reported in outcomes between any of the groups. Thus costs of complications is irrelevant
We include these costs for completeness.

VERSION 2 – REVIEW

REVIEWER	Aaron N Winn Medical College of Wisconsin, US
REVIEW RETURNED	21-Jan-2020

GENERAL COMMENTS	None- thank you for addressing my concerns.
---